# CROSS-STATE SELF-CONSTRAINT FOR FEATURE GENERALIZATION IN DEEP REINFORCEMENT LEARNING

## ABSTRACT

Representation learning on visualized input is an important yet challenging task for deep reinforcement learning (RL). The feature space learned from visualized input not only dominates the agent's generalization ability in new environments but also affect the data efficiency during training. To help the RL agent learn more general and discriminative representation among various states, we present cross-state self-constraint(CSSC), a novel technique that regularizes the representation feature space by comparing representation similarity across different pairs of state. Based on the implicit feedback between state and action from the agent's experience, this constraint helps reinforce the general feature recognition during the learning process and thus enhance the generalization to unseen environment. We test our proposed method on the OpenAI ProcGen benchmark and see significant improvement on generalization performance across most of Procgen games.

## 1 INTRODUCTION

Deep Reinforcement learning has achieved tremendous success on mastering video games(Mnih et al., 2015) and the game of GO(Silver et al., 2017). While training agent by using deep reinforcement learning algorithms, we usually assume that the agent could extract appropriate and effective features from different states and take actions accordingly. However, as more and more research works(Zhang et al. (2018), Song et al. (2019), Dabney et al. (2020)) have pointed out, even well-trained RL agents that learns from visualized input tend to memorizing spurious pattern rather than understanding the essential generic features of a given state. For example, an agent might pay more attention to the change of irrelevant background rather than noticing the obstacles or enemies(Song et al., 2019).

To improve generalization in the new environment, various kinds of regularization method like dropout(Farebrother et al., 2018) and data augmentation(Laskin et al., 2020) has been proposed and tested in combination with reinforcement learning. Conventional methods like dropout and batch-norm has been proven to be effective in supervised-learning, and for self-supervised learning like RL we see multiple related applications across various environments. Data augmentation like random crop(Laskin et al., 2020) or random convolution(Lee et al., 2019) have also been proposed recently and provide considerable generalization enhancement to the unseen levels of various tested environment(Tassa et al. (2018), Cobbe et al. (2018), Cobbe et al. (2020)). The agent is acting on multiple augmented views of the same input and learn from these prior injected data. However, modifying state information(injecting prior to the data) may be risky or even detrimental for representation learning because vital features may be altered or lost (ex: flipping state image might change the corresponding behavioral meaning, cropping the input image might lose critical features like the enemy position in the game).

To avoid losing informative features of the visualized input, we choose a different approach. As a human learner, we rarely depend on multiple augmented views of the same input to discriminate important or fictitious features. Instead, human learners try to recognize general patterns across multiple states and act accordingly. In other words, if the same action(or behavior) has been conducted by a well-trained agent in two different states, we would infer that the agent has conceived similar feature patterns in these states. For example, if one car stops for ninety seconds at two different intersections, we would guess that the car might be stopped by the red light at both places(Figure 1b).

From here we get the intuition about the relation between action(or behavior) and representation feature space learned by the agent: for a RL agent acting rationally across sequence of states, its behavior would be similar if it perceives similar critical patterns while acting differently if it perceives different patterns. Based on this intuition, we designed a novel constraint that performs regularization directly in the representation feature space of the learning agent. Our hypothesis for this constraint is simple: states in which the rational agent behaves identically should share more representation resemblance than states the agent behave differently.

We test this novel constraint in combination with Rainbow(Hessel et al., 2017), the state-of-the-art Q-learning method that combines numerous improvements with the original DQN model(Mnih et al., 2015). Inspired by the pair-wise structure used in BPR-opt(Rendle et al., 2012), we design the self-constraint based on the agent's behavior and utilize implicit feedback between positive and negative state pairs in the replay buffer. One thing worth noticing is that no change is needed for the underlying RL algorithm or model to adapt the proposed method, and it can be easily applied to other models with minimal effort(Figure 1a).

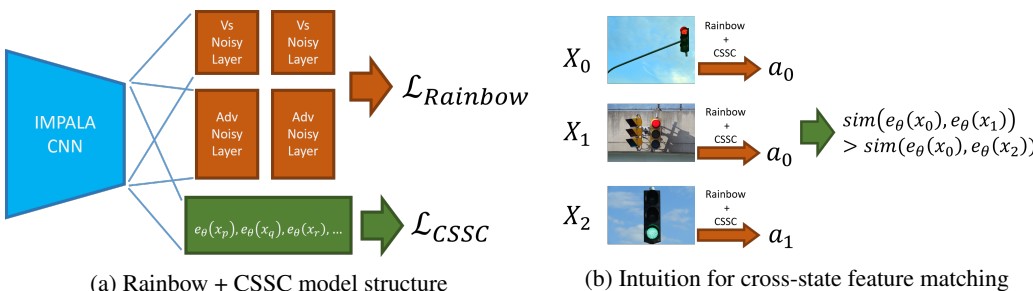

(a) Rainbow + CSSC model structure       (b) Intuition for cross-state feature matching

Figure 1: Model structure and CSSC concept

To measure the improvement on generalization to unseen environment levels, We test CSSC across 16 games on the OpenAI ProcGen benchmark(Cobbe et al., 2020). We see significant improvement on generalization capability for most ProcGen games in comparison with base Rainbow model.

We highlight the main contributions of CSSC as follows:

- Directly optimize feature generalization across various input states
- Requires no additional modification on the input data and the base model
- Enhance generalization and data efficiency on existing RL methods

## 2 RELATED WORK

### 2.1 AUXILIARY TASK FOR REPRESENTATION LEARNING IN RL

Representation Learning has been a vital part for deep RL algorithms. Even though the main goal of RL is to find the optimal value function, Mccallum & Ballard (1996) and Li et al. (2006) show that a representation specialized to this function may not be suitable for the sequence of value functions leading to it. On the other hand, Dabney et al. (2020) argue that the path toward optimal value function might be hindered by overfitting the representation to any intermediate value function during training. One popular and effective method to address this challenge is by using *auxiliary tasks*(Jaderberg et al. (2016), Bellemare et al. (2019)). Dabney et al. (2020) also propose new auxiliary tasks by considering the value-improvement path holistically.

### 2.2 REGULARIZATION ON RL

#### 2.2.1 ADDING STOCHASTICITY TO RL

Conventional practices like stochastic policy(Hausknecht & Stone, 2015), random starts(Mnih et al., 2015), sticky actions(Hausknecht & Stone, 2015) and frame skipping are wildly used for popular

tasks like Atari(Machado et al., 2017). By adding stochasticity to the environments during training and testing, we would prevent simple algorithms like *trajectory tree*(Kearns et al., 1999) and *brute*(Machado et al., 2017) from optimizing over open-loop sequences of actions without even considering the input states. However, as Zhang et al. (2018) has pointed out, injecting stochasticity to the maze-like environments does not necessarily prevent the deep RL agents from overfitting.

### 2.2.2 CONVENTIONAL REGULATIZATION ON RL

For deep supervised learning, regularizations has long been a major issue as generalization from training to test set is the major concern. Common regularization practices like L2 regularization, dropout, batch-normalization from supervised learning has also been tested in the reinforcement learning scenario. Cobbe et al. (2018) and Farebrother et al. (2018) show that dropout and L2 regularization improve generalization to test environments in Coinrun(Cobbe et al., 2018) and Atari 2600(Bellemare et al., 2013) respectively.

### 2.2.3 DATA AUGMENTATION ON RL

For supervised learning related to images or computer vision, data augmentation is a popular way to enhance performance by injecting useful priors on the training data. In recent years, we see numerous data augmentation methods for RL that improve generalization or data efficiency. Lee et al. (2019) introduce a randomized convolution network that randomly perturbs input features. Srinivas et al. (2020) utilize contrastive learning and show considerable performance gains in terms of sample-efficiency. Kostrikov et al. (2020) uses augmented data and weighted Q-functions to achieve state-of-the-art data-efficiency on the DMControl(Tassa et al., 2018). Laskin et al. (2020) investigate ten different data augmentation on RL and point out that random crop is the most effective on DMControl(Tassa et al., 2018)

## 3 BACKGROUND

CSSC is a general framework of cross-state representation regularization for RL. In principle, one can apply CSSC to other variation of DQN or policy-based models for discrete action-space environments. In this work we pick Rainbow (Hessel et al., 2017) and PPO (Schulman et al., 2017) as our base model to show that CSSC is compatible with the original Nature DQN(Mnih et al., 2015), its multiple improvements and policy-based algorithms. In the following subsections we will review Rainbow DQN and introduce the concept of implicit feedback in Bayesian Personalized Ranking (BPR) proposed by Rendle et al. (2012).

### 3.1 RAINBOW

In combination with a convolutional neural network as visualized input encoder, Deep Q network(Mnih et al., 2015) demonstrates that it is possible to use neural network as a function approximator that maps raw pixels to Q values. Since then, multiple improvements such as Double Q Learning(van Hasselt et al., 2015), Dueling Network Architecture(Wang et al., 2015), Prioritized Experience Replay(Schaul et al., 2015) and Noisy Networks(Fortunato et al., 2017) have been proposed. In addition, Bellemare et al. (2017) proposed the method of predicting a distribution over possible value support through the C51 algorithm. By combining all the above techniques into a single off-policy algorithm, Rainbow DQN showcases the state-of-the-art sample-efficiency on Atari Benchmarks. The resulting loss function for Rainbow DQN is as follows:

$$\mathcal{L}_{Rainbow} = D_{KL}(\Phi_z d_t^{(n)} || d_t) \tag{1}$$

where $\Phi_z$ is the projection onto fixed support $z$. $d_t^{(n)}$ and $d_t$ are the n-step target return distribution and the model-predicted return distribution respectively.

### 3.2 BPR-OPT

As opposed to explicit feedback like user ratings, implicit feedback in recommendation systems focuses on interactions like click or view between users and items. In implicit feedback system only

positive observations are available, and non-observed user-item pairs -e.g. a user has not view an item yet - are considered as negative observations. Instead of rating prediction, BPR-opt(Rendle et al., 2012) is designed for direct ranking optimization based on the implicit feedback between users and items. BPR extends the user preference from observed interaction pairs to non-observed data by ranking the preference of positive observation and negatived observation across the training data. The formulation of the maximum posterior estimator for the personalized ranking optimization (BPR-opt) can be written as follows:

$$\ln p(\Theta| >_u) = \sum_{(u,i,j)\in D_s} \ln \sigma(\hat{x}_{uij}) - \lambda_\Theta ||\Theta||^2 \tag{2}$$

where $\Theta$ represents the parameter vector of the base model. $D_s$ represent the batch samples from training data. $p(\Theta| >_u)$ is the posterior probability conditioned on latent preference structure $>_u$ for user $u$, $\lambda_\Theta$ is the model specific regularization parameter, $\sigma$ is the sigmoid function, and $\hat{x}_{uij}(\Theta)$ is the real-valued function model $\Theta$ which captures the preference relationship between user $u$, item $i$ and item $j$. Rendle et al. (2012) use $\hat{x}_{uij} := \hat{x}_{ui} - \hat{x}_{uj}$ to indicate that the user $u$ prefers item $i$ over item $j$.

## 4 METHOD

The main idea of CSSC is to extend the similarity ranking across various state representation based on our hypothesis: representations motivating identical behavior should share more similarity than those motivating different behavior. We name this method as "self-constraint" to emphasize that the state pairing is decided by the agent's behavior rather than pre-defined fixed labels. In the following subsection below, we introduce the following concepts to provide further explanation and insight: (i) definition of the behavior of an agent at a given state (ii) implicit feedback between state and behavior (iii) cross-state self-constraint as an auxiliary loss for representation regularization

### 4.1 BEHAVIOR DEFINITION

We describe a typical Markov Decision Process(MDP) as $(X, A, R, P, \gamma)$ with state space as $X$, action space as $A$, reward function as $R$, state transition function as $P$ and discount factor as $\gamma$. The agent would take an action $a_i$ at state $x_i$ and then being transited to $x_{i+1}$ by the environment with a given step reward $r_i$. Here we define the behavior set $B_i^n = \{(a_i, a_{i+1}, ..., a_{i+n-1})|a \in A\}$ as a set of action series of length n taken by the agent since state $x_i$. For $b_i^n \in B_i^n$ and $b_j^n \in B_j^n$, $b_i^n = b_j^n$ only if $a_{i+p} = a_{j+p}$ for $0 \le p \le n-1$. With this definition in mind, we can infer that if the agent conduct the behavior $b_i^2 = (left, fire)$ at state $x_i$, it would move left in state $x_i$ and fire in the next state $x_{i+1}$. In the following paragraph we coin names like unigram, bigram and trigram for behaviors of length one, two and three respectively.

### 4.2 IMPLICIT FEEDBACK IN REINFORCEMENT LEARNING

In the traditional deep Q-learning scenario, each state representation is highly correlated to neighboring states by the Bellman equation during the training process. However, observational overfitting could still happen as long as the representation of neighboring states can fit to specific sub-optimal version of Q-function indefinitely as mentioned in Dabney et al. (2020). Besides the base RL algorithm, we can view the interaction between state and behavior as an implicit feedback system by regarding states sharing same action as positive observation. The general feature among these positive observation would be reinforced if we compare their similarity with negative observation -e.g. states sharing different action. Through this "ranking" procedure the pair-wise relationship would be extended across non-neighboring states and therefore preventing observational overfitting to specific Q-function.

### 4.3 IMPLEMENTATION OF CROSS-STATE SELF-CONSTRAINT IN COMBINATION WITH RAINBOW AND PPO

For each sample of state triples $(x_p, x_q, x_r) \in X$ with $(b_p^n, b_q^n, b_r^n) \in B^n$ and $b_p^n = b_q^n \neq b_r^n$, we decompose the estimator $\hat{x}_{pqr}$ and define it as:

$$\hat{x}_{pqr} := \hat{x}_{pq} - \hat{x}_{pr} = e_\theta(x_p) \cdot e_\theta(x_q) - e_\theta(x_p) \cdot e_\theta(x_r) \tag{3}$$

where $e_\theta$ is the encoder function with weight $\theta$ that maps the pixel input to 1-D array feature vector with shape like [*element 0, element 1, ..., element n*] as the state representation. We directly perform inner-product on $e_\theta(x_p)$ and $e_\theta(x_q)$ to calculate the representation similarity $\hat{x}_{pq}$ between $x_p$ and $x_q$. Please note that the encoder function is the same part of the original base model used for feature extraction on visualized input. For more details about neural network structures used in this paper, please refer to Figure 7 and Figure 8. To sample these state triplets without modifying the base DQN or PPO algorithm, we collect these state triples in the same batch of transitions used to calculate Bellman loss or policy loss. For policy-based algorithms and DQN without prioritized replay (Schaul et al., 2015), we pair up the state triples for each transition in the sample and calculate the CSSC loss as follows:

$$\mathcal{L}_{CSSC} = -\frac{1}{N_{D_s}} \sum_{(p,q,r) \in D_s} \ln \sigma(\hat{x}_{pqr}) \tag{4}$$

where $D_s$ represents the sample batch of size $N_{D_s}$ from the replay buffer and $\sigma$ is the sigmoid function. Then, we train the base algorithm in combination with the auxiliary CSSC loss as follows:

$$\mathcal{L}_{total} = \mathcal{L}_{base} + \beta_{cssc} \cdot \mathcal{L}_{CSSC} \tag{5}$$

where $\beta_{cssc}$ is the hyperparameter to control the contribution of CSSC during training. For pseudocode of PPO with unigram CSSC, please refer to Algorithm 2 in the appendix.

In the case of Rainbow with CSSC, we design the loss function in combination with the importance sampling (IP) weight as follows:

$$\widehat{\mathcal{L}_{total}} = (\widehat{\mathcal{L}_{Rainbow}} \bigoplus (\beta_{cssc} \cdot \widehat{\mathcal{L}_{CSSC}})) \odot \widehat{W_{IP}} \tag{6}$$

where $\bigoplus$ and $\odot$ are element-wise add and multiplication respectively. For pseudocode of Rainbow with unigram CSSC, please refer to Algorithm 1 in the appendix.

We find that setting $\beta_{cssc}$ to 0.01 works quite well in most cases, so we stick with this setting for all the experiment conducted in the next section.

## 5 EXPERIMENT

Our primary goal for CSSC is to enhance the generalization capability of RL algorithm to unseen levels that share similar mechanism. Fortunately, OpenAI ProcGen(Cobbe et al., 2020) presents a collection of game-like environments where the training and testing environments differs in visual appearance and structure. Therefore, we evaluate CSSC in three different ways:(i) generalization improvement on 12 easy-mode games and 8 hard-mode games with Rainbow on OpenAI Proc-Gen (ii) generalization improvement on 16 easy-mode games with PPO on OpenAI ProcGen (iii) performance improvement on 32 games with Rainbow on Gym Atari (iv) visualization on the representation space learned with CSSC.

Table 1: Mean normalized score of Rainbow in OpenAI ProcGen Easy Mode

| Mode | Rainbow | Rainbow + uni-CSSC | Rainbow + bi-CSSC |
|---|---|---|---|
| easy(train)@25M (12 Games) | 0.2862 | 0.3832 | 0.4030 |
| easy(test)@25M (12 Games) | 0.1018 | 0.1670 | 0.1892 |
| Improve. for train(%) | 0.0 | 33.89 | 40.81 |
| Improve. for test(%) | 0.0 | 64.05 | 85.85 |

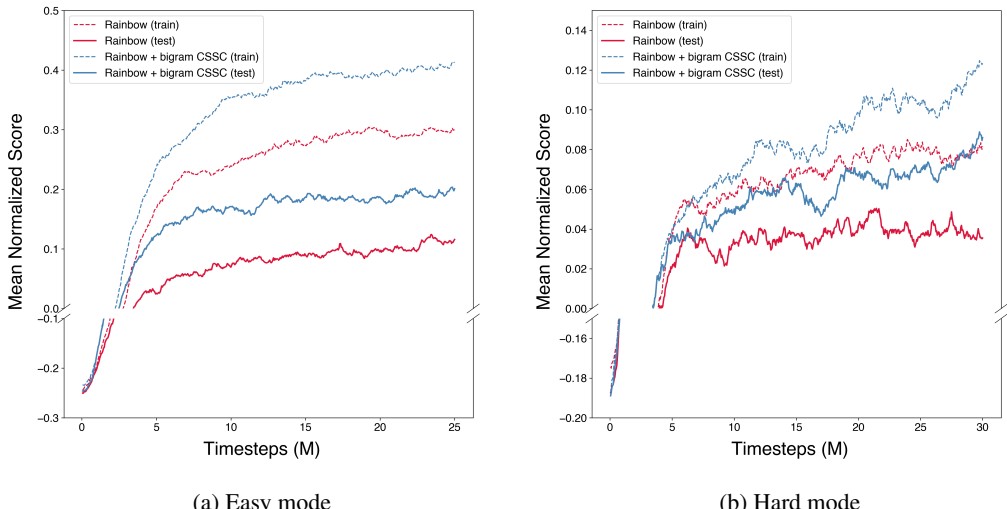

(a) Easy mode                                    (b) Hard mode

Figure 2: Mean normalized score curve on OpenAI ProcGen environment. We normalized the episodic return of each game by the constants from Cobbe et al. (2020) and report the mean score. Every curve is smoothed with an exponential moving average of 0.95 to improve readability.

Table 2: Mean normalized scores of Rainbow bigram-CSSC

| Mode | Rainbow | Rainbow + bi-CSSC | Improvement(%) |
|---|---|---|---|
| easy(train)@25M (12 Games) | 0.2862 | 0.4030 | 40.81 |
| easy(test)@25M (12 Games) | 0.1018 | 0.1892 | 85.85 |
| hard(train)@30M (8 Games) | 0.1085 | 0.1699 | 56.63 |
| hard(test)@30M (8 Games) | 0.0512 | 0.1199 | 133.94 |

## 5.1 ENVIRONMENT AND MODEL SETTING

For the following experiments, we use the IMPALA CNN architecture recommended by Cobbe et al. (2020) for the Rainbow and PPO model on ProcGen benchmark. For the Gym Atari benchmark, we stay with the Nature CNN as mentioned in Hessel et al. (2017) for the Rainbow model. All the experiments in this paper are conducted in the single-agent setting. For all hyperparameter used on ProcGen and Atari benchmark, please refer to Table 3 in the Appendix. We use "unigram CSSC" or "bigram CSSC" to indicate the behavior length used to pair up the state triples for CSSC.

## 5.2 GENERALIZATION ON PROCGEN IN EASY MODE WITH RAINBOW

As Cobbe et al. (2020) have suggested, we conduct the training of 25 million timesteps on 200 training levels and evaluate the generalization improvement on the full distribution of testing levels across 12 ProcGen games. We test both the unigram and bigram CSSC in comparison with vanilla Rainbow and see significant improvement on testing performance. To evaluate the mean normalized score for bigram and unigram CSSC across 12 games, we follow the normalization method used in Cobbe et al. (2020) and show the result in Figure 2a and Figure 12a respectively. We also display the learning curve of bigram-CSSC in Figure 3 and unigram-CSSC in Figure 13. In the following we summarize the main findings below:

- In supervised learning, posing regularization or constraint to the model would usually improve its testing performance on unseen data at the expense of hurting training performance. However, we see that the proposed self-constraint improves both training and testing performance across most of the 12 game tested.

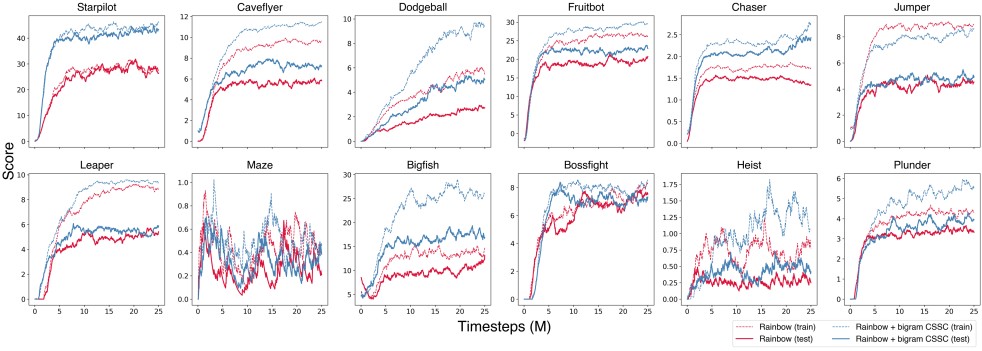

Figure 3: Learning Curve of Rainbow with bigram CSSC in 12 ProcGen Games. We report the raw episodic return for both training and testing. All final scores are listed in Table 5. Every curve is smoothed with an exponential moving average of 0.95 to improve readability.

- For games like Starpilot, Chaser and Bigfish, we see that the testing performance is even better than the training performance of vanilla Rainbow.
- For the mean normalized score in Table 1, Bigram-CSSC and Unigram-CSSC bring 85% and 64% improvement respectively on the testing performance in comparison with base Rainbow model.

## 5.3 GENERALIZATION ON PROCGEN IN HARD MODE WITH RAINBOW

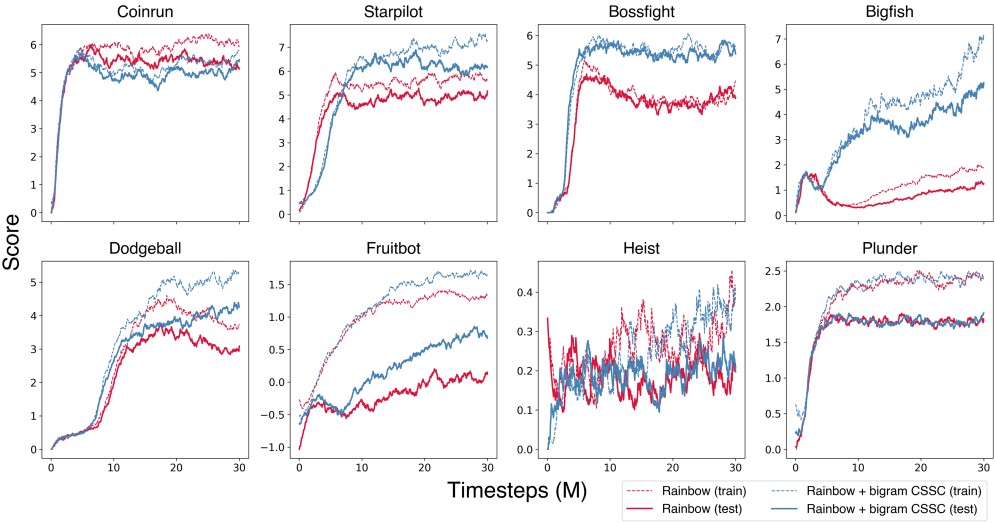

Figure 4: Learning Curve of Rainbow with bigram CSSC in 8 ProcGen Games. We report the mean raw episodic return for training and testing. Every mean return is shown across 3 seeds and all final scores are listed in Table 6. Every curve is smoothed with an exponential moving average of 0.95 to improve readability.

To further explore the generalization capability in more complicated and challenging tasks, we examine the generalization improvement brought by CSSC on 8 ProcGen games. We use 500 training levels for 30 million timesteps of training across 3 seeds and evaluate generalization on the full distribution of testing levels. We test the bigram CSSC in comparison with the normal version of Rainbow across three different seeds and see significant improvement on testing performance. The learning curve of bigram CSSC is summarized in Figure 4. To evaluate the mean normalized score for bigram CSSC across 8 games in hard mode, we also follow the normalization method used in

Cobbe et al. (2020) and show the result in Figure 2b. In the following we summarize the main findings below:

- As shown in Figure4, CSSC substantially improves both training and testing performance in Bigfish, Dodgeball, Starpilot, Fruitbot and Bossfight.

- In particular, we see nearly 4x performance jump in Bigfish at 30M timestep in comparison with the base Rainbow model. The mean episodic return at 30M timestep of bigram CSSC is even higher than that of PPO at 200M timestep as shown in Figure 4 of Cobbe et al. (2020).

- In the case of Heist (a puzzle-solving task in maze-like layout), coinrun(a 2-D scroll platform game) and Plunder(a challenging shooting game), the gain from CSSC is not as obvious because all these games require careful manipulation and planning. Add constraint that modifies state representation directly would be risky or even detrimental for the learning process of the base RL algorithm.

### 5.4 GENERALIZATION ON PROCGEN IN EASY MODE WITH PPO

We test the CSSC with PPO on all the 16 games of the Procgen benchmark in Easy mode. We conduct the training of 25 million timesteps across 3 seeds on 200 training levels and evaluate the generalization improvement on the full distribution of testing levels across all 16 ProcGen games. From the normalized learning curve shown in Figure 5a we can tell that CSSC helps reduce the gap between training and testing performance. For detailed final scores and learning curves, please refer to Table 7 and Figure 14, 15 in the Appendix.

### 5.5 PERFORMANCE ON GYM ATARI WITH RAINBOW

We take Gym Atari benchmark (Brockman et al., 2016) as our second set of environments to measure the effectiveness of CSSC. Even though this classic benchmark does not explicitly split the training levels out of the training levels in all environments, we still see significant improvement on 23 out of 38 games tested as shown in Figure 5b. For training hyperparameters and scoreboard on Gym Atari benchmark, please refer to Table 3 and 8 in the Appendix.

### 5.6 VISUALIZATION OF REPRESENTATION EMBEDDING

To give a more tangible explanation about the effectiveness of CSSC, we plot the representation distribution in the embedding feature space across 4096 states from the replay buffer. We first perform dimension reduction on these state embeddings using principal decomposition analysis and label each embedding point with the index of corresponding action. Here we show the embedding space of both Rainbow and bigram-CSSC models trained to play the Bigfish game. In figure 6 we can see that states of the same action are more clustered in bigram-CSSC than that of the vanilla Rainbow model. We believe this is the possible reason behind the significant improvement on generalization as shown in Figure 4. For representation visualization in other ProcGen games, please refer to A.4.

## 6 CONCLUSION

In this paper we propose a novel regularization on state representation learning based on the connection between agent behavior and the visual input. Our hypothesis is derived from the observation that the behavior of a rational agent would have certain relationship with general cross-state features or patterns. We see significant improvement on generalization brought by the proposed cross-state self-constraint(CSSC) on most of the games in OpenAI ProcGen benchmark. The connection between behavior and conceived visual input can be considered as some kind of "motivation" that acts as a decisive factor behind the learned policy. It worth further study to better understand the derivation and transformation of agent's "motivation" during the Markov Decision Process, and we believe the concept proposed in this work would facilitate more research in this direction.

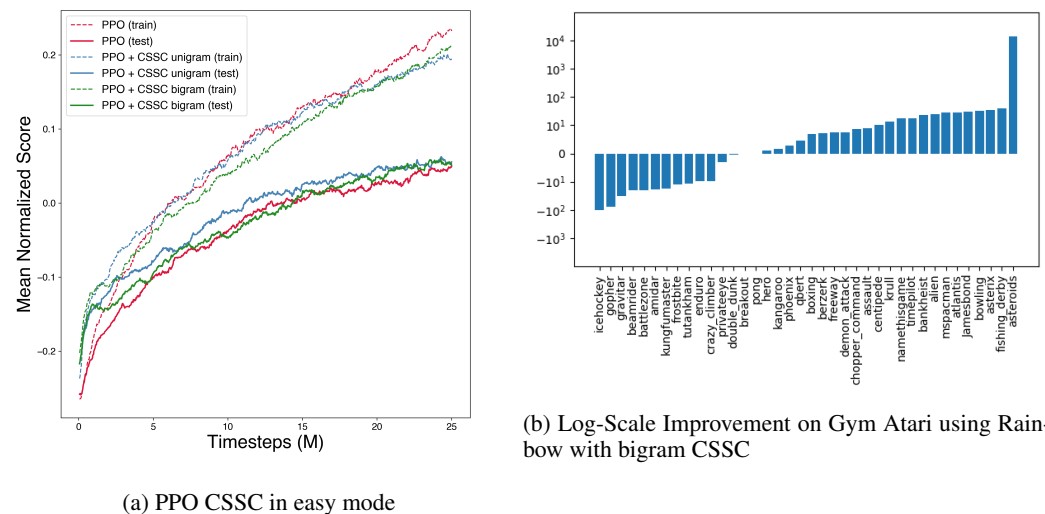

(a) PPO CSSC in easy mode

(b) Log-Scale Improvement on Gym Atari using Rainbow with bigram CSSC

Figure 5: Left: Mean normalized score curve on OpenAI ProcGen environment. We normalized the episodic return of each game by the constants from Cobbe et al. (2020) and report the mean score. Every curve is averaged across 3 seeds and is smoothed with an exponential moving average of 0.95 to improve readability. Right: the performance improvement of bigram CSSC with Rainbow on 38 games from the Gym Atari benchmark. All scores are recorded using the same random seed.

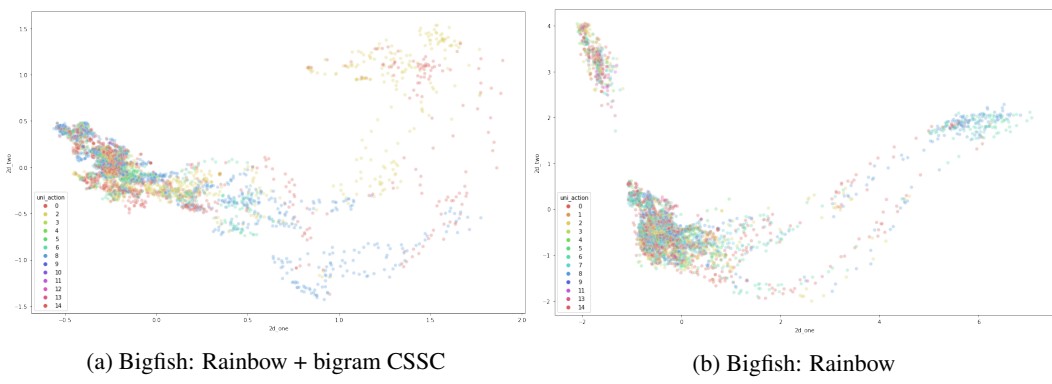

(a) Bigfish: Rainbow + bigram CSSC

(b) Bigfish: Rainbow

Figure 6: Representation embedding projected to 2D space by PCA. We use the same 4096 state frames to extract representation using the bigram-CSSC and vanilla Rainbow model and display the distribution across all state representation.

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

# A  APPENDIX

## A.1  NEURAL NETWORK ARCHITECTURE

- Rainbow IMPALA network

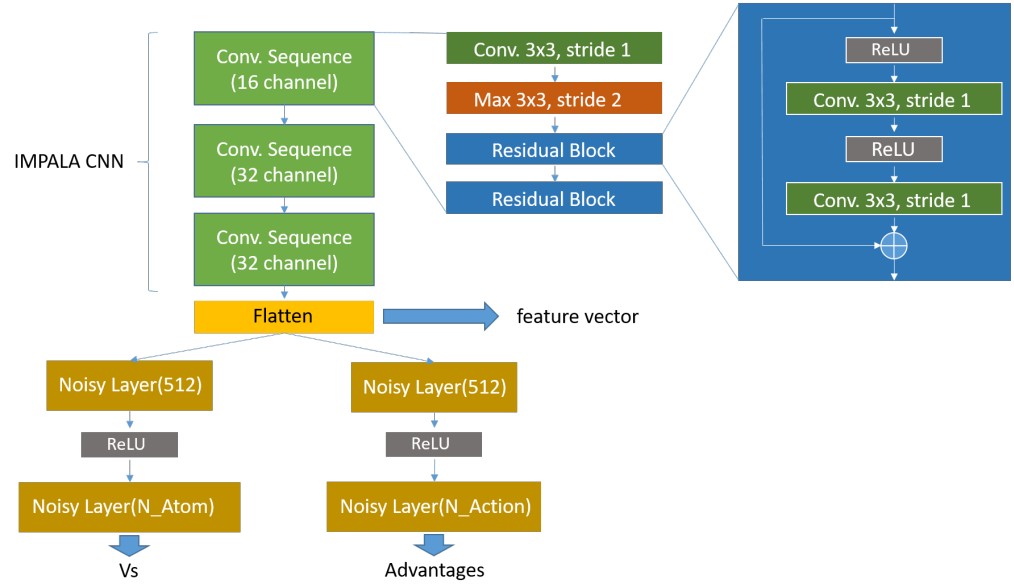

Figure 7: Neural network architecture used for the Rainbow CSSC experiment. The output of the feature vector for each input state is taken as the state embedding for CSSC loss.

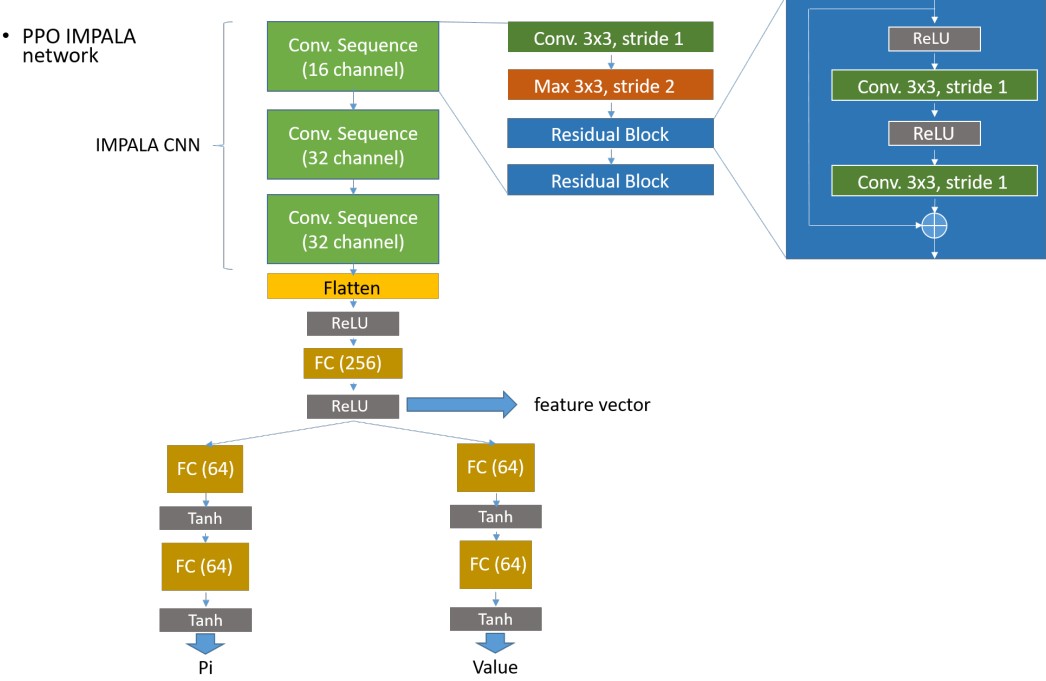

Figure 8: Neural network architecture used for the PPO CSSC experiment. The output of the feature vector for each input state is taken as the state embedding for CSSC loss.

A.2 HYPERPARAMTERS AND MODEL SETTING FOR OPENAI PROCGEN AND GYM ATARI BENCHMARK

Table 3: Hyperparameters of Rainbow for OpenAI ProcGen

| Env | ProcGen(Easy) | ProcGen(Hard) | Atari |
|---|---|---|---|
| Use stacked Frames | False | False | True |
| Image size(Channel, Height, Width) | (3, 64, 64) | (3, 64, 64) | (4, 84, 84) |
| CNN network | IMPALA CNN | IMPALA CNN | Nature CNN |
| state feature vector size | 2048 | 2048 | 3136 |
| Training levels | 200 | 500 | None |
| Training start Level | 123 | 123 | None |
| Testing levels | 0 | 0 | None |
| Testing start level | 0 | 0 | None |
| Batch size | 128 | 128 | 32 |
| CSSC type | unigram, bigram | bigram | bigram |
| CSSC constant $\beta_{CSSC}$ | 0.01 | 0.01 | 0.01 |
| Multi step return | 3 | 3 | 3 |
| gamma | 0.99 | 0.99 | 0.99 |
| Learning rate | 0.00025 | 0.00025 | 0.0000625 |
| Replay buffer size | 500000 | 500000 | 1000000 |
| Replay alpha | 0.5 | 0.5 | 0.5 |
| Replay beta start | 0.4 | 0.4 | 0.4 |
| Minimum replay size for sampling | 20000 | 20000 | 80000 |
| Replay frequency | 16 | 16 | 4 |
| Target network update period | 8000 | 8000 | 8000 |
| Total timesteps | 25M | 30M | 50M |

Table 4: Hyperparameters of PPO for OpenAI ProcGen

| Env | ProcGen(Easy) |
|---|---|
| Use stacked Frames | False |
| Image size(Channel, Height, Width) | (3, 64, 64) |
| state feature vector size | 256 |
| Training levels | 200 |
| Training start Level | 0 |
| Testing levels | 0 |
| Testing start level | 0 |
| Batch size | 2048 |
| number of epoch | 3 |
| vf coef | 0.25 |
| entropy coef | 0.01 |
| gamma | 0.999 |
| lambda | 0.95 |
| clip range | 0.2 |
| clip range vf | 0.2 |
| max grad norm | 0.5 |
| CSSC type | unigram, bigram, trigram |
| CSSC constant $\beta_{CSSC}$ | 0.01 |
| Learning rate | 0.0005 |
| rollout buffer size | 16384 |
| number of envs | 64 |
| steps per env | 256 |
| Total timesteps | 25M |

## A.3 SCOREBOARDS ON OPENAI PROCGEN

Table 5: Rainbow: Easy mode scores evaluated after 25M timesteps of training on 1 seed.

| Game | Train | | | Test | | |
|------|-------|-------|-------|-------|-------|-------|
| | Rainbow | Unigram | Bigram | Rainbow | Unigram | Bigram |
| Starpilot | 27.79 | 43.75 | **46.13** | 26.37 | 39.72 | **43.40** |
| Caveflyer | 9.70 | 10.63 | **11.43** | 5.83 | 7.22 | **7.23** |
| Dodgeball | 5.46 | 9.08 | **9.48** | 2.74 | 4.25 | **5.04** |
| Fruitbot | 26.25 | 29.27 | **29.68** | 20.75 | **23.81** | 22.91 |
| Chaser | 1.71 | 2.33 | **2.74** | 1.34 | 2.07 | **2.43** |
| Jumper | **8.96** | 8.73 | 8.62 | 4.54 | **4.99** | 4.97 |
| Leaper | 8.81 | **9.58** | 9.39 | 5.42 | 5.83 | **5.88** |
| Maze | 0.41 | **0.71** | 0.45 | 0.21 | **0.50** | 0.38 |
| Bigfish | 14.80 | **26.60** | 26.04 | 13.16 | 14.87 | **16.91** |
| Bossfight | 8.26 | 8.16 | **8.57** | **7.52** | 6.90 | 7.32 |
| Heist | 0.77 | 0.77 | **1.06** | 0.26 | 0.34 | **0.42** |
| Plunder | 4.34 | 4.51 | **5.58** | 3.32 | 2.75 | **3.95** |

Table 6: Rainbow: Hard mode scores evaluated after 30M timesteps of training on 3 seeds.

| Game | Train (mean ± std) | | Test (mean ± std) | |
|------|--------------------|-------|--------------------|-------|
| | Rainbow | Bigram | Rainbow | Bigram |
| Coinrun | **5.99±0.18** | 5.78±0.28 | 5.13±0.15 | **5.46±0.37** |
| Starpilot | 5.63±0.23 | **7.30±0.27** | 5.17±0.15 | **6.16±0.26** |
| Bossfight | 4.42±0.43 | **5.64±0.52** | 3.89±0.24 | **5.38±0.28** |
| Bigfish | 1.88±0.41 | **7.22±4.27** | 1.24±0.38 | **5.26±3.28** |
| Dodgeball | 3.74±0.15 | **5.23±0.28** | 3.09±0.21 | **4.26±0.08** |
| Fruitbot | 1.33±0.14 | **1.65±0.19** | 0.12±0.60 | **0.69±0.22** |
| Heist | 0.38±0.06 | **0.39±0.03** | **0.21±0.01** | 0.20±0.03 |
| Plunder | **2.43±0.08** | 2.38±0.07 | 1.81±0.03 | **1.91±0.04** |

Table 7: PPO: Easy mode scores evaluated after 25M timesteps of training on 3 seeds.

| Game | Train (mean ± std) | | | Test (mean ± std) | | |
|------|--------------------|---------|--------|--------------------|---------|--------|
| | PPO | Unigram | Bigram | PPO | Unigram | Bigram |
| Bigfish | 3.96 ±1.35 | **6.77 ±1.86** | 6.30 ±1.93 | 1.45 ±0.30 | **3.29 ±1.45** | 2.33 ±0.66 |
| Bossfight | **8.36 ±0.24** | 7.21 ±0.51 | 7.51 ±0.38 | **7.64 ±0.14** | 6.79 ±0.42 | 6.38 ±0.61 |
| caveflyer | 1.59 ±0.29 | **2.53 ±0.48** | 1.97 ±0.87 | 0.76 ±0.40 | **1.78 ±0.33** | 1.34 ±0.27 |
| Chaser | **1.98 ±0.23** | 1.93 ±0.14 | 1.77 ±0.14 | **1.73 ±0.08** | 1.63 ±0.11 | 1.51 ±0.07 |
| Climber | 2.63 ±0.07 | **3.22 ±0.82** | 2.52 ±0.47 | 1.42 ±0.17 | 1.53 ±0.24 | **1.70 ±0.19** |
| Coinrun | **7.34 ±0.67** | 6.84 ±1.03 | 6.69 ±0.52 | **6.35 ±0.34** | 6.16 ±0.62 | 5.12 ±0.36 |
| Dodgeball | **1.37 ±0.30** | 1.33 ±0.20 | 1.31 ±0.03 | **0.89 ±0.10** | 0.88 ±0.07 | 0.80 ±0.02 |
| Fruitbot | 29.27±0.51 | 29.86±0.43 | **30.05±0.38** | 24.99±0.71 | **27.00±0.45** | 25.63±0.13 |
| Heist | 1.38 ±0.21 | 1.77 ±0.03 | **2.42 ±0.45** | 0.27 ±0.11 | **0.39 ±0.16** | 0.35 ±0.05 |
| Jumper | **7.90 ±0.09** | 7.23 ±0.20 | 6.99 ±0.27 | **4.16 ±0.15** | 4.04 ±0.51 | 3.98 ±0.60 |
| Leaper | 4.38 ±0.33 | 4.58 ±0.27 | **4.86 ±0.34** | 4.39 ±0.18 | 4.03 ±0.12 | **4.75 ±0.35** |
| Maze | **6.10 ±0.77** | 4.75 ±0.97 | 5.79 ±0.42 | 2.55 ±0.36 | **2.70 ±0.37** | 2.58 ±0.15 |
| Miner | **7.37 ±1.62** | 2.22 ±0.58 | 3.59 ±0.45 | **1.80 ±0.45** | 1.24 ±0.06 | 1.21 ±0.08 |
| Ninja | 3.76 ±0.54 | 3.76 ±0.26 | **4.40 ±0.37** | 2.86 ±0.07 | 2.53 ±0.36 | **3.63 ±0.29** |
| Plunder | 3.37 ±0.36 | **3.71 ±0.36** | 3.21 ±0.26 | 2.96 ±0.34 | **3.17 ±0.16** | 2.94 ±0.24 |
| Starpilot | 27.19±2.42 | 26.19±1.77 | **28.41±2.23** | 19.82±0.85 | 22.12±1.87 | **22.28±1.37** |

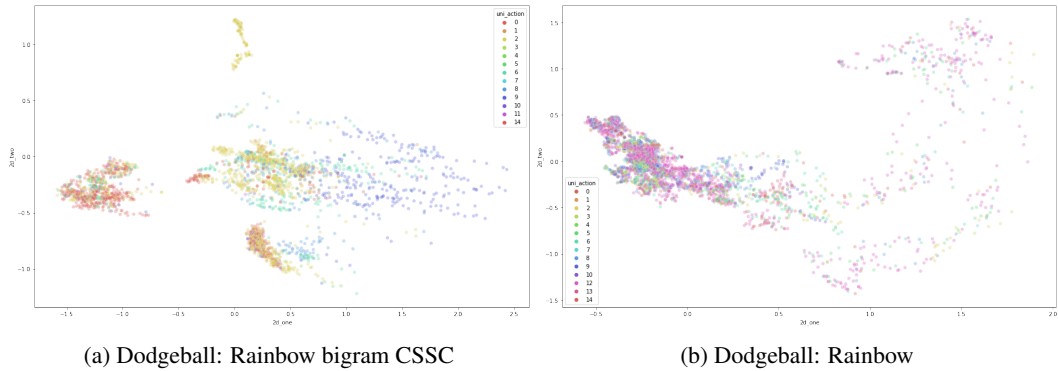

(a) Dodgeball: Rainbow bigram CSSC        (b) Dodgeball: Rainbow

Figure 9: Dodgeball: representation embedding projected to 2D space by PCA

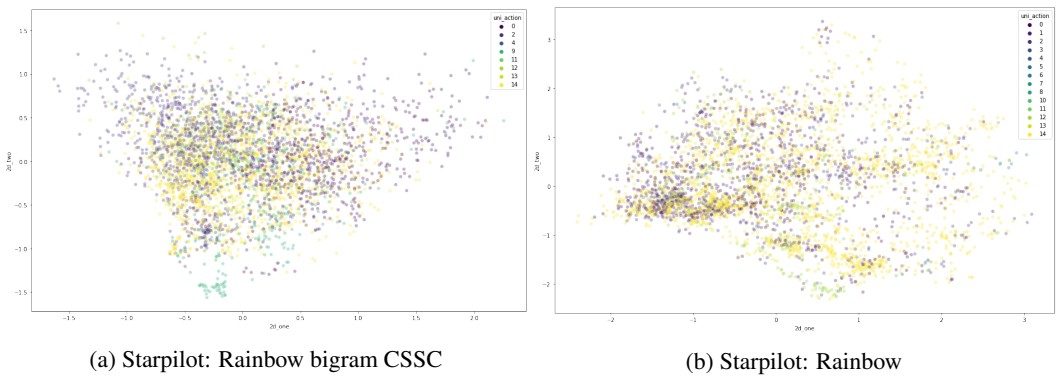

(a) Starpilot: Rainbow bigram CSSC        (b) Starpilot: Rainbow

Figure 10: Starpilot: representation embedding projected to 2D space by PCA

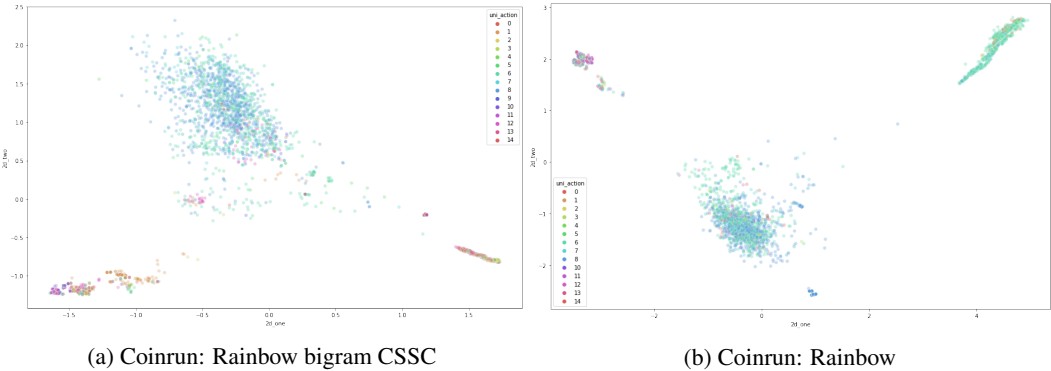

(a) Coinrun: Rainbow bigram CSSC        (b) Coinrun: Rainbow

Figure 11: Coinrun: representation embedding projected to 2D space by PCA

A.4    VISUALIZATION OF REPRESENTATION EMBEDDING ON OPENAI PROCGEN GAMES

A.5    GENERALIZATION ON PROCGEN IN EASY MODE FOR RAINBOW UNIGRAM-CSSC

A.6    GENERALIZATION ON PROCGEN IN EASY MODE FOR PPO WITH CSSC

A.7    PSEUDOCODE FOR CSSC EXPERIMENT

A.8    EVALUATION OF BIGRAM CSSC WITH RAINBOW ON GYM ATARI BENCHMARK

---

**Algorithm 1** Rainbow + CSSC (unigram)

---

1: **Input:** minibatch size $k$, multi-step return $n$, replay period $K$ and buffer size $N$, exponents $\alpha$ and $\beta$, gamma $\gamma$, CSSC coefficient $\beta_{cssc}$, total-timesteps $T$.

2: Initialize replay memory $\mathcal{H} = \emptyset$, feature extractor $\theta$, return distribution predictor $\Theta$, $p_1 = 1$

3: Observe state $s_0$ and choose action $a_0 \sim \pi_\theta(s_0)$

4: **for** $t = 1$ to $T$ **do**

5:     Observe state $s_t$, reward $r_t^{(n)}$, done flag $d_t$, new state $s_{t+n}$

6:     Store transition $j_t = (s_t, a_t, r_t^{(n)}, d_t, s_{t+n})$ in $\mathcal{H}$ with maximal priority $p_t = \max_{i<t} p_i$

7:     **if** $t \mod K \equiv 0$ **then**

8:         Sample batch of transition $J = [j_0, j_1, ..., j_k | j \sim P(j) = p_j^\alpha / \Sigma_i p_i^\alpha]$

9:         Compute importance-sampling weight: $\widehat{W} = [w_0, w_1, ..., w_k | w_j = (N \cdot P(j))^{-\beta} / \max_i w_i]$

10:         Predict state feature vector $e_\theta(s_j), e_\theta(s_{j+n})$ and return distribution $d_j, d_j^{(n)} = \Theta(e_\theta(s_j)), \Theta(e_\theta(s_{j+n}))$ for each transition in the batch

11:         Compute $\widehat{\mathcal{L}_{Rainbow}} = [l_0, l_1, ..., l_k | l_j = D_{KL}(\Phi_z d_j^{(n)} || d_j)]$ using distribution projection function $\Phi_z$

12:         Update transition priority $p_j \leftarrow l_j$ for each transition in the batch

13:         Sample positive pair $(s_p, s_q), p \neq q$ with $j_p, j_q \in J$ and behavior $b_p^1 = b_q^1$ for each transition $j_p$ in J

14:         Sample negative pair $(s_p, s_r), p \neq r$ with $j_p, j_r \in J$ and behavior $b_p^1 \neq b_r^1$ for each transition $j_p$ in J

15:         Compute $\widehat{\mathcal{L}_{CSSC}} = [\hat{s}_{0_{p_0 q_0 r_0}}, ..., \hat{s}_{k_{p_k q_k r_k}} | \hat{s}_{j_{p_j q_j r_j}} := -1.0 \times \sigma(e_\theta(s_{p_j}) \cdot e_\theta(s_{q_j}) - e_\theta(s_{p_j}) \cdot e_\theta(s_{r_j}))]$

16:         Compute $\widehat{\mathcal{L}_{total}} = (\widehat{\mathcal{L}_{Rainbow}} \bigoplus (\beta_{CSSC} \cdot \widehat{\mathcal{L}_{CSSC}})) \odot \widehat{W}$ where $\bigoplus$ and $\odot$ are element-wise add and multiplication respectively

17:         Fit $\Theta$ and $\theta$ with Adam optimizer to minimize $\widehat{\mathcal{L}_{total}}$.mean()

18:     **end if**

19: **end for**

---

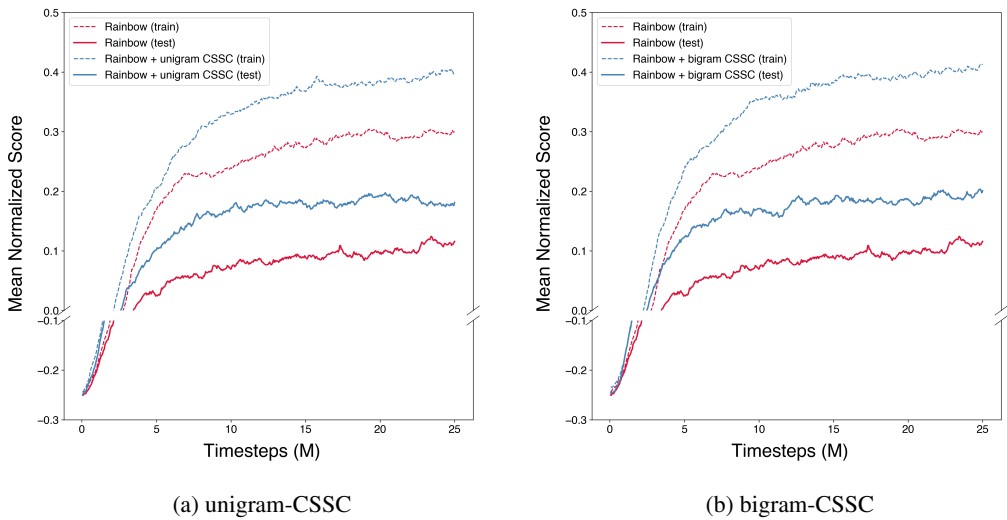

(a) unigram-CSSC

(b) bigram-CSSC

Figure 12: Mean normalized score on OpenAI ProcGen environment in easy mode. Every curve is smoothed with an exponential moving average of 0.95 to improve readability.

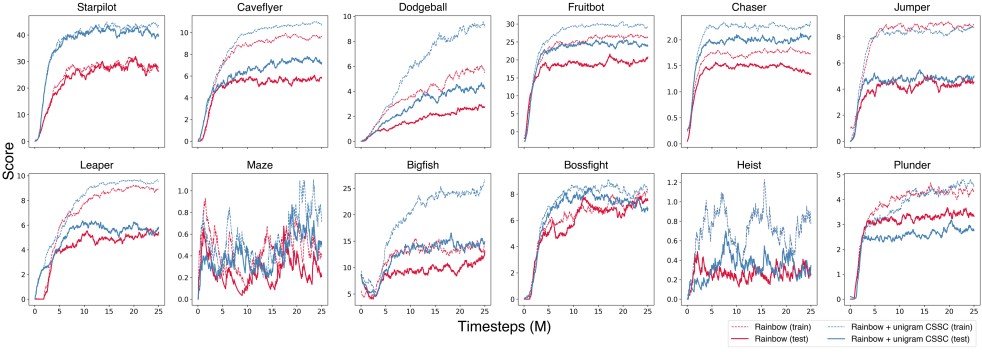

Figure 13: Learning Curve of Rainbow with unigram CSSC in 12 ProcGen Games. We report the raw episodic return for both training and testing. Every curve is smoothed with an exponential moving average of 0.95 to improve readability.

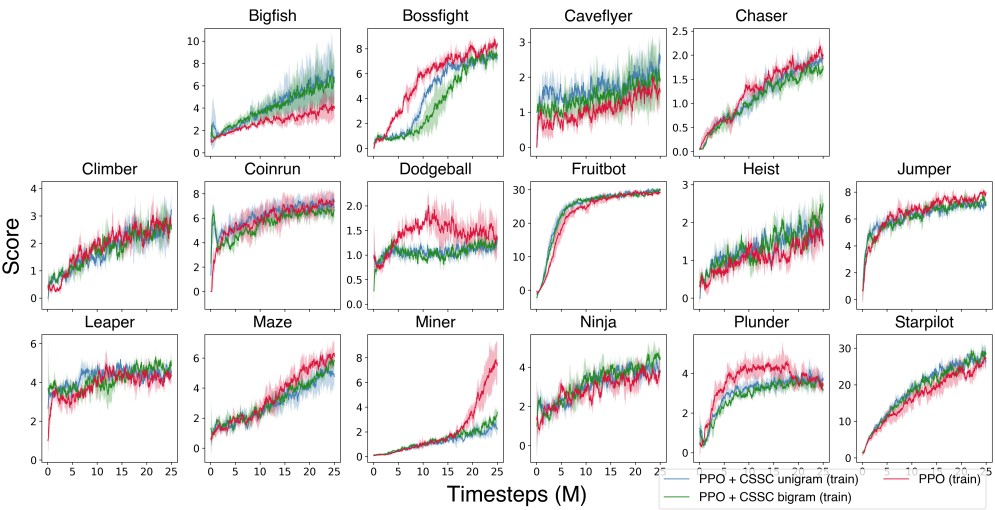

Figure 14: Learning Curves of PPO in 16 ProcGen Games on training levels. Every curve is averaged across 3 seeds and smoothed with an exponential moving average of 0.95 to improve readability.

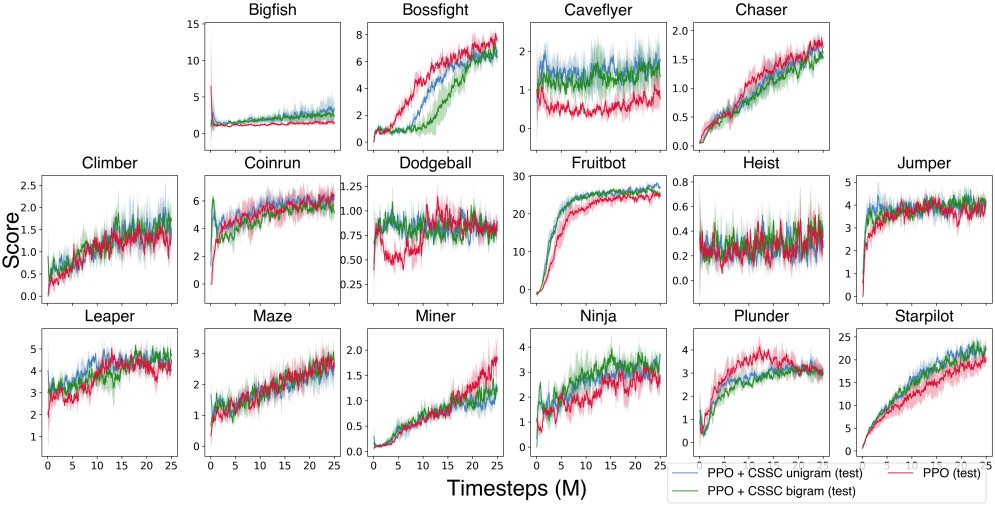

Figure 15: Learning Curves of PPO with CSSC in 16 ProcGen Games on testing levels. Every curve is averaged across 3 seeds and smoothed with an exponential moving average of 0.95 to improve readability.

---

**Algorithm 2** PPO + CSSC (unigram)

---

1: **Input:** minibatch size $k$, CSSC coefficient $\beta_{cssc}$, total-timesteps $T$, clip_range
2: Initialize rollout buffer $\mathcal{H} = \emptyset$, feature extractor $\theta$, policy parameters $\Theta$, value function parameters $\phi$
3: **for** $t = 0$ to $T$ **do**
4:      Collect set of trajectories $D_k = \{\mathcal{T}_i\}$ by running policy $\pi_k = \pi(\theta_k)$ in the environment
5:      Compute advantages estimates $\hat{A}_t$ using current value function $V_\phi$
6:      Store transition $j_t = (s_t, a_t, r_t, d_t, \hat{A}_t, log(\pi(\theta_k)))$ in $\mathcal{H}$
7:      **if** rollout buffer $\mathcal{H}$ is full **then**
8:          **for** each batch in $\mathcal{H}$ **do**
9:              Predict state feature vector $e_\theta(s_j)$, log_prob, values using $\pi_\theta$ and $V_\phi$ for transitions in the batch
10:              Compute ratio = exp(log_prob - batch.old_log_prob)
11:              Compute policy_loss = batch.advantages $\cdot$ clamp(ratio, 1 - clip_range, 1 + clip_range)
12:              Compute $\mathcal{L}_{policy}$ = policy_loss.mean()
13:              Sample positive pair $(s_p, s_q), p \neq q$ with $j_p, j_q \in J$ and behavior $b_p^1 = b_q^1$ for each transition $j_p$ in batch transitions J
14:              Sample negative pair $(s_p, s_r), p \neq r$ with $j_p, j_r \in J$ and behavior $b_p^1 \neq b_r^1$ for each transition $j_p$ in batch transitions J
15:              Compute cssc_loss = $[\hat{s}_{0 p_0 q_0 r_0}, ..., \hat{s}_{k p_k q_k r_k} | \hat{s}_{j_{p_j q_j r_j}} := -1.0 \times \sigma(e_\theta(s_{p_j}) \cdot e_\theta(s_{q_j}) - e_\theta(s_{p_j}) \cdot e_\theta(s_{r_j}))]$
16:              Compute $\mathcal{L}_{CSSC}$ = cssc_loss.mean()
17:              Fit $\Theta$, $\theta$ and $\phi$ with Adam optimizer to minimize $\mathcal{L}_{total} = \mathcal{L}_{Rainbow} + \beta_{cssc} \cdot \mathcal{L}_{CSSC}$
18:          **end for**
19:      **end if**
20: **end for**

---

Table 8: Gym Atari scores evaluated after 50M timesteps of training on 1 seed.

| | Train (mean) | |
| --- | --- | --- |
| Game | Rainbow | Bigram |
| alien | 3,738.3 | 4,659.5 |
| amidar | 1,951.2 | 1,579.2 |
| assault | 18,292.2 | 19,690.2 |
| asterix | 480,431.1 | 647,559.0 |
| asteroids | 2,041.0 | 296,158.1 |
| atlantis | 2,419,949.7 | 3,090,829.1 |
| bankheist | 1,292.9 | 1,595.3 |
| battlezone | 77,924.6 | 62,216.0 |
| beamrider | 35,120.4 | 27,937.9 |
| berzerk | 2,974.7 | 3,130.8 |
| bowling | 55.5 | 74.0 |
| boxing | 94.8 | 99.6 |
| breakout | 417.9 | 413.5 |
| centipede | 18,103.3 | 19,926.0 |
| chopper command | 28,468.0 | 30,509.5 |
| crazy climber | 243,610.0 | 221,021.5 |
| demon attack | 506,716.2 | 535,951.2 |
| double dunk | -22.6 | -22.8 |
| enduro | 7069.1 | 6391.6 |
| fishing derby | 36.18 | 50.4 |
| freeway | 32.0 | 33.8 |
| frostbite | 11688.2 | 10268.2 |
| gopher | 151,072.1 | 34,115.3 |
| gravitar | 2,084.5 | 1,426.3 |
| hero | 33,534.5 | 33,965.5 |
| icehockey | 4.9 | -0.125 |
| Jamesbond | 36,417.5 | 47,388.5 |
| kangaroo | 14,525.0 | 14,740.5 |
| krull | 7,843.5 | 8,913.0 |
| kungfumaster | 51,319.0 | 42,107.0 |
| mspacman | 3,468.2 | 4,420.7 |
| namethisgame | 11,525.2 | 13,574.9 |
| phoenix | 479,379.5 | 488,284.2 |
| pong | 20.7 | 20.9 |
| privateeye | 99.5 | 100.0 |
| qbert | 26,394.9 | 27,159.0 |
| timepilot | 22930.0 | 27053.0 |
| tutankham | 170.4 | 150.7 |

