# OpenReview forum: "Cross-State Self-Constraint for Feature Generalization in Deep Reinforcement Learning"
_ICLR.cc/2021/Conference — Reject_

### Official Review · AnonReviewer3 · 2020-10-16
**Nice idea but stronger baselines are missing**

**Rating:** 5
**Confidence:** 4

**Review:**

Summary:
The paper proposes to introduce an auxiliary contrastive loss to improve generalization on procedurally generated environments. In particular, it encourages latent representations of states to be more similar when those states lead the current agent to take the same sequence of future actions.

Positive:
(+)  Clear idea & description
(+) Simple, but novel idea
(+) Positive experimental results

Weaknesses:
(-) Insufficient comparison to baselines
(-) Only one set of benchmark environments

Recommendation: 5
I'm currently recommending (weak) rejection but would change my score based on additional experimental results.
In particular: As noted by the authors, prior work has shown that "standard" regularization techniques like L2 (weight decay), Batchnorm or Dropout can improve performance on both training and testing. As the authors introduce a novel type of regularization technique, I believe it is important to not only compare against a "vanilla" (i.e. unregularized) agent, but agents utilizing various simpler regularization techniques.
Their technique would not even need to outperform agents which are, for example, regularized by weight decay, but I'd find it sufficient to show that CSSC can further improve the performance of such already regularized agents.

Minor point:
A second set of other procedurally generated environments, other than ProcGen, would help show that the method is not overfit to ProcGen.

Other remarks (no need to answer, didn't have impact on evaluation):
- I found section 4.2 hard to understand, maybe it could be reformulated?
- Page 5, first line: I can't believe that e_\theta maps to a 1d feature vector (i.e. a scalar)?
- Equation (3): x_j should probably be x_r

---

> ### Author Response · Authors · 2020-11-23
> **Modifications related to issues mentioned by reviewers**
>
> Thanks for your insightful and informative feedback. We’ve added some modifications in the paper related to issues mentioned above:
>
> (1)	A second set of other procedurally generated environments: We have added evaluation about CSSC on Gym Atari benchmark in section 5.5 of the revised paper. We find that CSSC brings measurable improvement on 23 out of 38 games tested. Please refer to the revised paper for further details.
>
> (2)	$e_\theta$ maps to a 1d feature vector (i.e. a scalar)? Sorry for using the ambiguous word. Here we actually mean 1-D “array” feature vectors having the shape like [ element 0, element 1, …, element n]. In the case of Rainbow with IMPALA CNN network this 1-D array feature vectors have 2048 elements. In the case of PPO we follow the IMPALA CNN network structure of the original Procgen Paper and extract 1-D array feature vector of 256 elements for every input frame.
>
> We would also study the effectiveness of CSSC in combination with other regularization or generalization method (ex: dropout, batch norm, data augmentation…) in the future. We also add experiment about CSSC with PPO in the revised paper. Please let us know if you have any question about the revised paper and thanks again for your feedback.

---

> > ### Comment · AnonReviewer3 · 2020-11-24
> > **Thank you**
> >
> > Dear Authors,
> >
> > thank you for the updated paper and the provided clarifications.
> >
> > While the added experimental results are a great addition, my main concern about the lack of fair comparison against other, widely available regularisation methods, was not addressed yet, so I will keep my score.
> >
> > I believe the paper provides a very interesting idea but requires those baselines in order to make a convincing argument.
> > As such, I would strongly encourage the authors to include them, either for ICLR in case of acceptance or in a future submission.

---

### Official Review · AnonReviewer1 · 2020-10-23
**Empirical good results, some clarifications/improvements needed**

**Rating:** 6
**Confidence:** 3

**Review:**

This paper tackles the problem of representation learning from visualized input. The paper presents "cross-state self-constraint(CSSC)", a technique for regularizing the representation feature space by favoring representation similarity (scalar product between representations) between representations when the agent behaves similarly. The approach is tested with deep RL on the OpenAI ProcGen benchmark.

The method developped in this paper seems to provide interesting empirical results. The motivation of bringing in all cases the representation closer when the agent takes the same actions is not necessarily obvious because very different reasons might lead to the same sequence of actions, but this empirical study seems to shows the interest of this idea.

My main concern is related to the following remarks/questions that I believe require some clarifications:

- In equation 3, what does j stands for?
- Can $\hat x_{pqr}$ be negative? In that case, what happens with the loss?
- n sometimes correspond to "a set of action series of length $n$" and sometimes to the batch size. Usually a batch of tuples is selected randomly in the replay memory (without notion of sequence). What does n corresponds to in practice in the experiments?
- In table 1 and 2: How many seeds are used? Why is the standard deviation not given?
- No information on the neural network architecture seems to be given. What is the structure of the abstract representation?

Some typos (the 12 game tested, Rainbwo model)

---

> ### Author Response · Authors · 2020-11-23
> **Modifications related to issues mentioned by reviewers**
>
> Thanks for your informative feedback. We’ve added some modifications in the paper related to issues mentioned above:
>
> (1)	In equation 3, what does j stands for?: Thanks for pointing that out. It should be "r" and we have fixed that up in the revised paper.
>
> (2)	Can $\hat x_{pqr}$ be negative?: Yes, $\hat x_{pqr}$ can be negative. Because y = LogSigmoid(x) would approximate y = x for x < 0 and y = 0 for x > 0, the loss term would actually work in the cast for negative $\hat x_{pqr}$. A negative $\hat x_{pqr}$ indicates that the proposed constraint is violated and thus we use LogSigmoid to generate the corresponding loss term.
>
> (3)	What does n corresponds to in practice in the experiments? We use n to indicate the length of action sequence and $N_{D_s}$ for the batch size in the revised paper. Thanks for pointing that out.
>
> (4)	How many seeds are used? Why is the standard deviation not given? We have added the number of seed and the std of each run in the updated paper. Please refer to the Table 5,6,7 in Appendix A.3 for more details
>
> (5)	What is the structure of the abstract representation? We have added the illustration about the neural network architecture in Appendix A.1. Please refer to Appendix A.1. for more details.
>
> Please let us know if you have any question about the revised paper.

---

> > ### Comment · AnonReviewer1 · 2020-11-23
> > **Thanks for the clarifications and additions**
> >
> > Thanks for your answers and the modifications done in the paper accordingly. I'm willing to increase my score to 6. As stated in my initial review and even though the idea developed in the paper seems to be interesting given the empirical study, I do not give a score higher than 6 because the motivation of the work is not necessarily very clear. Intuitive explanations of why this idea is of interest and/or a theoretical explanation would be needed to be a clear accept.

---

### Official Review · AnonReviewer2 · 2020-10-27
**Interesting idea but needs more work**

**Rating:** 5
**Confidence:** 4

**Review:**

Summary

This paper proposes a new method for learning representations of images with the goal of improving generalization in RL. The key idea is to regularize the learned feature space by forcing embeddings of states followed by the same action (or sequence of actions) to be more similar than embeddings of states followed by different actions. The method is evaluated on the Procgen and achieves superior performance relative to Rainbow, a standard RL algorithm.


Strengths

I do like the core idea of this paper and I think it makes sense intuitively. I also think the paper is trying to solve an important problem in RL which has been neglected until recently. In addition, the proposed method is simple and achieves much better generalization than Rainbow. However, I think the paper needs more work to be ready for publication for the reasons laid out below.


Weaknesses

The main weakness of this paper is the lack of comparisons with previously proposed methods for improving generalization in RL. While the results are strong relative to Rainbow, no other baselines are included. I suggest adding comparisons to other methods which have recently shown good results on Procgen such as PPO (used as a baseline in the paper that introduced Procgen from Cobbe et al. 2020), DrAC (Raileanu et al. 2020), ITER (Igl et al. 2020), and DRIML (Mazoure et al 2020).

In addition, various choices made by the authors make it difficult to compare with prior art.  For the hard mode, you only train the agents for 30m steps while the original paper trains them for 200m steps in order to achieve decent performance. You also only show results on 12 or 8 of the games instead of all 16. Following the same training practices and evaluation metrics as the ones proposed in Cobbe et al. 2020 would make comparisons with previous and future work easier and more transparent.

You mention using a 1-dimensional feature vector for the state representation. That seems very low and I don’t understand how it could capture all the relevant details in an observation. Does the method work with higher dimensional representations? Additional experiments and discussion to motivate this choice would be useful.

As I understand, the embeddings are constrained using the CSSC at the same time as the policy is being learned. But at the beginning of training, the actions are random so constraining representations based on the actions taken doesn’t seem ideal. Wouldn’t this slow down training because the CSSC loss forces the “wrong constraint” at the beginning and thus lead to poor representations which would then make it harder to learn a good policy? Can you include some experiments and discussion about how the performance varies with the coefficient for the CSSC loss.

Have you tried using trigrams? It seems like using bigrams is better than unigrams so it would be useful to see whether adding more actions continues to improve performance or not.

While I liked the visualization of the learned embeddings, I think it would be more insightful to compare the embeddings of observations from train and test (colored by their actions) in order to understand how well the structure in the embedding space generalizes to unseen observations. Do you see similar structure in the embeddings of test observations as in the train ones?

I found the related work section to be rather disconnected. The authors mention relevant works but do not discuss them in relation to the proposed approach. I suggest the authors discuss in more detail the limitations of current methods and explain why (and when) the proposed approach might be preferable. In addition, the section is missing a number of closely related papers that address representation learning for improving generalization in RL such as Farebrother et al. 2018, Igl et al. 2019 and 2020, Lee et al. 2020, Srinivas et al. 2020, Laskin et al. 2020, Roy et al. 2020, Mazoure et al. 2020, Zhang et al. 2020, Raileanu et al. 2020.

I think the clarity of the paper can be significantly improved. For example, at the end of the intro you mention that there are significant improvements for most Procgen games, but it would be better to be more precise and mention exactly on how many of them it is better and even by how much (i.e. outperforms on X out of Y games and achieves z% better test performance on the entire benchmark). The following sentence can also be more specific: “ Inspired by the pair-wise structure used in BPR-opt(Rendle et al., 2012), we design the self-constraint based on our hypothesis and utilize implicit feedback between positive and negative state pairs in the replay buffer“ by replacing the word “hypothesis” with something more descriptive. I would also more gently introduce the relation to BPR-opt for readers who don’t know what that is -- this is the first mention of BPR and has no explanation of what it is, including what the acronym stands for. There are lots of other places in the paper where clarity needs to be improved.


Minor Points

Some of the formatting seems off e.g. there is no space before starting a parenthesis.

Sometimes you use “representation feature space” which seems redundant. Using “representation space” or “feature space” would be more concise.

The use of “visualized input” instead of “visual input” sounds strange to me and it’s not a common phrasing in this field as far as I know.

Typos:
In equation 3, I believe it should be x_r instead of x_j
“Has conceived” instead of “has conceive” in the intro
“Novel constraint that performs” instead of “...that perform” in the intro
“Rational agent behaves” instead of “...behave” in intro
Some letters are capitalized after a comma.
“The most effective” instead of “...effect” in related work
“Demonstrates” instead of “demonstrates” in 3.1
Many others…


Recommendation

While the proposed method is simple, sensible, and appears effective, the paper needs significant improvements as I explained above. Thus, I cannot recommend it for publication at this stage but I encourage the authors to continue working on it.

---

> ### Author Response · Authors · 2020-11-23
> **Modifications and discussions about issues mentioned by reviewers**
>
> Thanks for your detailed and elaborated feedback. We deeply appreciate the insight and suggestions mentioned in the review. We’ve added some modifications in the paper related to issues mentioned above:
>
> (1)	The lack of comparisons with previously proposed methods: We’ve added the variance for experiments on Rainbow and PPO. Please refer to Table 6 and 7 in the Appendix A.3 for more details. We would also study other methods like DrAC, ITER and DRIML and try to add comparisons with them in the next few weeks.
>
> (2)	Various choices made by the authors make it difficult to compare with prior art: The Rainbow model we used in this work is trained in single-agent environment, which needs substantial wall-clock time to finish just one training process (about 5 days for 50 M steps). In the original ProcGen paper the authors mentioned about using 4 rainbow agents to speed up the training process but they did not reveal the corresponding code. We would try to build distributed training framework like that to speed up the training process in the following weeks.
>
> (3)	Using a 1-dimensional feature vector for the state representation: Sorry for using the ambiguous word. Here we actually mean 1-D “array” feature vectors having the shape like [ element 0, element 1, …, element n]. In the case of Rainbow with IMPALA CNN network this 1-D array feature vectors have 2048 elements. In the case of PPO we follow the IMPALA CNN network structure mentioned in the original Procgen Paper and extract 1-D array feature vector of 256 elements for every input frame.
>
> (4)	“wrong constraint” at the beginning: We do have considered using a variational CSSC coefficient by changing the coefficient during the training process (like that of epsilon-greedy exploration). To our surprise, the constant coefficient version of CSSC does provides measurable improvement across various games without too much fine-tuning. We would also study the effectiveness of CSSC using the variational coefficient in the future.
>
> (5)	Have you tried using trigrams? : We have tested trigram CSSC on some ProcGen games but it does not outperform unigram or bigram in most cases. The valid number of positive pair in trigram is low because very few states would share identical action sequence of length 3 and thus having minimal effect on the learning process.
>
> (6)	Do you see similar structure in the embeddings of test observations as in the train ones? : Great suggestion! We would conduct similar analysis on testing levels in the next few days if time permits.
>
> We would further polish up the paper and add more analysis about the proposed method in the next few weeks. Please let us know if you have any other question about the revised paper and thanks again for your informative feedback.

---

> > ### Comment · AnonReviewer2 · 2020-11-23
> > **Reviewer Response**
> >
> > Thank you for your response and for updating the paper. It was valuable to hear your insights about the CSSC coefficient and the use of trigrams. I also appreciated the additional experiments with PPO and on Atari.
> >
> > It seems like the gains of CSSC over PPO are rather marginal. Do you have an intuition of why there is such a big difference between the boost in performance you get using CSSC with Rainbow v. PPO?
> >
> > Given these additional experiments and clarifications, I will increase my score to 5.
> >
> > I also suggest reporting results over at least 5 seeds given the well known variance of RL algorithms.
> >
> > However, I believe the main limitation of this paper is the lack of comparisons with additional baselines (e.g., DrAC, DRIML, ITER etc.). As other reviewers have mentioned, CSSC seems orthogonal to other types of regularization so it would be useful to see how it compares with them as well as whether it can provide further benefits on top of other methods for generalization in RL. For these reasons, I cannot whole-heartedly recommend the paper for acceptance in its current form, but I encourage the authors to use this feedback to strengthen the paper.

---

> > > ### Author Response · Authors · 2020-11-24
> > > **Extra imformation about PPO and other baselines**
> > >
> > > Thanks again for your extra feedback about the updated paper.
> > >
> > > The performance related to CSSC with PPO still have plenty of room for improvement due to the lack of fine-tuning and tweaking about the PPO implementation. We adopt the PPO implementation on stable-baselines 3 (https://github.com/DLR-RM/stable-baselines3) but find out that the same set of hyperparameters used in the original ProcGen paper (which use https://github.com/openai/baselines) could not bring corresponding results on the PPO of stable-baselines 3. We are still studying the difference between these two open-sourced implementation and would keep evaluating the effectiveness of CSSC on top of SB3.
> > >
> > > We would also study other baselines like (e.g., DrAC, DRIML, ITER etc.) and evaluate the possibility of combining CSSC with other method as well. In comparison with other method, CSSC leaves the original neural network structure intact and brings minimal computational burden to the algorithm. Therefore we are excited to test CSSC with other generalization methods.

---

### Official Review · AnonReviewer4 · 2020-10-29
**Feature generalization imposed via an additional loss**

**Rating:** 5
**Confidence:** 4

**Review:**

The authors modify the Rainbow loss through an additional loss based on a cross-state similarity. The notion of similarity is called in the paper "cross-state self-constraint" - CSSC. The CSSC loss looks very simple and can be applied with other RL algorithms. The loss is defined in terms of an embedding e of the visual input into a 1-dim space. Given three states x_p, x_q, and x_r, their similarity is defined as the scalar product of e(x_p) and e(x_q) minus the scalar product of e(x_q) and e(x_r). It would be useful if the authors include pseudocode how the loss is exactly computed for a given batch of samples. My understanding of the description is that for a given batch the authors generate triples with the same action set and then take as an auxiliary loss the weighted sum of the logarithms of sigmoids of similarities of these triples.

The results on the ProcGen suite of environments are reported in Tables 1 and 2 and they look promising. Unfortunately, the results are reported without information about the variance. Also, for some environments, e.g. for Chaser, the final numbers are weak compared to the ProcGen paper.  The inclusion of a more systematic comparison with PPO (using numbers from the ProcGen paper) would be helpful.

The visualizations look interesting, though I am not sure how the authors back their claim that  "in figure 5 we can see that states of the same action are more clustered in bigram-CSSC". Can you include some numerical information about clustering and/or add some extra annotations to figure 5?

My understanding of the paper is that the standard IMPALA architecture was equipped with an additional "embedding head" (I am inferring it from Figure 1). Though I cannot find a precise description of this architecture change, in particular how large are the embedding vectors.

The method looks simple and interesting, though the paper in my opinion needs a substantial re-write before publication at ICLR.

---

> ### Author Response · Authors · 2020-11-23
> **Modifications related to issues mentioned by reviewers**
>
> Thank you for your insightful feedback. We’ve added some modifications in the paper related to issues mentioned above:
>
> (1)	Visual input into a 1-dim space: Sorry for using the ambiguous word. Here we actually mean 1-D “array” feature vectors having the shape like [ element 0, element 1, …, element n]. In the case of Rainbow with IMPALA CNN network the 1-D array feature vectors have 2048 elements. In the case of PPO we also follow the IMPALA CNN network structure mentioned in the original Procgen Paper which extract 1-D array feature vector of 256 elements for every input frame.
>
> (2)	It would be useful if the authors include pseudo code: We’ve added the pseudo code in the Appendix A.7.
>
> (3)	Results are reported without information about the variance: We’ve added the variance for experiments on Rainbow and PPO. Please refer to Table 6 and 7 in the Appendix A.3 for more details.
>
> (4)	The inclusion of a more systematic comparison with PPO: We’ve added experimental result for PPO with CSSC. Please refer to section 5.4 for more details.
>
> (5)	Though I cannot find a precise description of this architecture change, in particular how large are the embedding vectors: We’ve added the illustration about the neural network structure used in this paper and the state embedding size in appendix A.1 and A.2.
>
> We will also try to add more clustering and extra annotations to figure 5 if time permits. We are willing to give further explanation if you have any question about the modified paper.

---

### Comment · ~Rishabh_Agarwal2 · 2021-01-14
**Very related to Behavioral Similarity Embeddings for Generalization (concurrent work at ICLR'21)**

Interesting work! The similarity of states based on action is very similar CSSC is very similar to the [policy similarity metric](agarwl.github.io/pse) (with a small discount factor such as 0.1 or smaller).  We didn't use ProcGen for evaluation in our work, but it's exciting to see that a similar approach to ours works well on ProcGen too!

Paper: https://openreview.net/forum?id=qda7-sVg84

---

### Decision · Program_Chairs · 2021-01-07
**Final Decision**

**Decision:**

Reject

**Comment:**

I thank the authors for their submission and participation in the author response period. The updated experiments are appreciated. However, after discussion all reviewers unanimously agree that the paper is not ready for publication and encourage resubmission to another venue. In particular, R2 and R3 have raised concerns regarding additional widely available baselines that need to be evaluated against and that the rebuttal has not addressed. I agree with this assessment, and thus recommend rejection.